# Comparative Assessment and External Validation of Hepatic Steatosis Formulae in a Community-Based Setting

**DOI:** 10.3390/jcm9092851

**Published:** 2020-09-03

**Authors:** Tae Yang Jung, Myung Sub Kim, Hyun Pyo Hong, Kyung A Kang, Dae Won Jun

**Affiliations:** 1Department of Internal Medicine, Hanyang University College of Medicine, Seoul 04763, Korea; megatonsun@naver.com; 2Department of Radiology, Kangbuk Samsung Hospital, Sungkyunkwan University School of Medicine, Seoul 03181, Korea; smmgkim84@naver.com (M.S.K.); summersonrad@naver.com (H.P.H.)

**Keywords:** non-alcoholic fatty liver disease, fatty liver, validation study, population groups, magnetic resonance imaging

## Abstract

Several hepatic steatosis formulae have been validated in various cohorts using ultrasonography. However, none of these studies has been validated in a community-based setting using the gold standard method. Thus, the aim of this study was to externally validate hepatic steatosis formulae in community-based settings using magnetic resonance imaging (MRI). A total of 1301 community-based health checkup subjects who underwent liver fat quantification with MRI were enrolled in this study. Diagnostic performance was assessed using the area under the receiver operating characteristic curve (AUROC). Non-alcoholic fatty liver disease (NAFLD) liver fat score showed the highest diagnostic performance with an AUROC of 0.72, followed by Framingham steatosis index (0.70), hepatic steatosis index (HSI, 0.69), ZJU index (0.69), and fatty liver index (FLI, 0.68). There were considerable gray zones in three fatty liver prediction models using two cutoffs (FLI, 28.9%; HSI, 48.9%; and ZJU index, 53.6%). The diagnostic performance of NAFLD liver fat score for detecting steatosis was comparable to that of ultrasonography. The diagnostic agreement was 72.7% between NAFLD liver fat score and 70.9% between ultrasound and MRI. In conclusion, the NAFLD liver fat score showed the best diagnostic performance for detecting hepatic steatosis. Its diagnostic performance was comparable to that of ultrasonography in a community-based setting.

## 1. Introduction

Non-alcoholic fatty liver disease (NAFLD) is the most common cause of chronic liver disease. The prevalence of NAFLD has increased along with that of diabetes, obesity, and metabolic syndrome [1]. Recognizing fatty liver is the first step in the screening for those with a high risk of steatohepatitis in the general population. Although hepatic fibrosis is a more serious and clinically significant disorder that may progress to cirrhosis, early detection of simple steatosis is also an important and promising field from a public health system view.

Various prediction models have been developed for hepatic steatosis using routine biochemical test results and anthropometric parameters as simple screening tools. Some prediction models for hepatic steatosis have persistently demonstrated acceptable diagnostic performances in several cohorts. However, most studies were performed using ultrasonography (USG) [2,3,4,5,6,7,8]. Although USG is widely used as a screening tool for detecting the presence of fatty liver in the general population, the quality of the USG is highly operator dependent because it is based on a subjective assessment of liver echogenicity [9]. Liver biopsy is the “gold standard” method to assess hepatic steatosis. However, it is impossible to perform it on a large scale for the general population. Magnetic resonance imaging (MRI) can provide a reliable estimation of fatty liver. It is now considered as “the next best method” as compared with liver biopsy [10]. To date, only five studies have validated prediction models of hepatic steatosis using gold standard [11,12,13,14,15]. One study used liver biopsy [11] and the other four used proton magnetic resonance spectroscopy (^1^H-MRS) [12,13,14,15]. However, all five studies were performed on selected populations with small sample sizes.

The diagnostic performance, especially positive predictive value (PPV) and negative predictive value (NPV), critically depends on the prevalence of the disease [16]. Therefore, external validation studies for “average-risk group” using gold standard method are needed. To the best of our knowledge, there is no appropriate external validation study for existing fatty liver prediction models using MRI in the general population. Thus, the aim of this study was to comparatively assess and externally validate existing hepatic steatosis formulae in a community-based setting.

## 2. Materials and Methods

### 2.1. Study Population

A retrospective analysis was performed using health examination data collected by a medical examination center. The population of this study was composed of 2149 Korean adults who underwent a medical examination and an MRI in Kangbuk Samsung Hospital Healthcare Screening Center (Seoul and Suwon, Korea) between January 2015 and May 2018. According to the Occupational Safety and Health Act, it is mandatory for companies with more than 40 employees to provide empolyee benefits for annual or biannual medical examinations in Korea. MRI scans are voluntary chosen by employees and the charge is paid by their employer. Over 90% of the study population were employees and their family members. This study was approved by the Kangbuk Samsung Hospital Institutional Review Board (IRB No. KBSMC 2019-12-002). The clinical protocol was registered with the Korean Clinical Research Information Service with a registration number of KCT0004645.

### 2.2. Inclusion and Exclusion Criteria

The study population included 2149 subjects aged 18 years or older who visited Kangbuk Samsung Hospital’s Health Screening Center with their liver fat content measured using MRI. We excluded subjects with viral hepatitis B (*n* = 167) or C (*n* = 12), subjects with chronic liver disease caused by significant alcohol consumption (>20 g/day, *n* = 642) [17,18], and subjects with suspected chronic liver disease or liver cirrhosis on USG (*n* = 27) in order to exclude chronic liver disease with unknown etiology such as autoimmune hepatitis, primary biliary cirrhosis, and hemochromatosis (Figure 1).

### 2.3. Imaging Assessment

MRI examinations were performed using 1.5 T scanners (Signa HDxT; GE Healthcare, Milwaukee, WI, USA, in Seoul; and Optima 360 Advance, GE Healthcare, Milwaukee, WI, USA, in Suwon). The protocol sequence included coronal and axial T2-weighted sequences, an axial T1-weighted sequence, a free-breathing diffusion-weighted sequence, and a dual-echo in- and opposed-phase T1-weighted gradient recalled echo sequence. No intravenous contrast was administered.

### 2.4. Fat Fraction Quantification

A dual-echo chemical shift imaging technique was used for quantitative assessment of hepatic steatosis. For each image pair, a circular region-of-interest was placed in the right liver lobe by three experienced abdominal radiologists. For large vessels, motion artifacts or focal liver lesions observed on maps were carefully avoided.

The fat fraction was calculated using the following formula, considering net signal in liver on opposed-phase images in comparison with in-phase images: fat fraction = (in-phase image signal intensity—opposed-phase image signal intensity)/(2 × in-phase image signal intensity) × 100. The degree of steatosis was quantified as follows: normal (<5%), mild (5–11%), moderate (11–17%), or severe (>17%) [19].

### 2.5. Anthropometric Measurements

Height and body weight were measured by trained nurses. Waist circumference was measured from the midpoint between immediately below the rib cage and the iliac crest. Sitting blood pressure (BP) was measured using an automatic tonometer by placing the cuff at the level of the heart. We administered a questionnaire to evaluate the frequency of alcohol intake and alcohol consumption on a single occasion.

### 2.6. Definitions

Diabetes was diagnosed for patients with a history of diabetes, having HbA1C of 6.5 or higher, or having a fasting glucose level higher than 126 mg/dL according to the WHO criteria. Hypertension was defined for subjects with systolic BP ≥140 mm Hg, diastolic BP ≥90 mm Hg, or patients already receiving hypertension medication. Subjects were considered to have metabolic syndrome if three of the following five conditions were met: (1) waist circumference ≥90 cm for men, ≥80 cm for women in accordance with the International Obesity Task Force criteria for the Asian-Pacific population; (2) triglycerides (TG) ≥150 mg/dL; (3) high-density lipoprotein cholesterol (HDL-C) <40 mg/dL for men or <50 mg/dL for women; (4) BP ≥130 mm Hg (systolic) or ≥85 mmHg (diastolic); and (5) fasting glucose ≥100 mg/dL. NAFLD was defined as fat deposition in the liver ≥5% on MRI in the absence of viral hepatitis, significant alcohol consumption (>20 g/day) [17,18] and other chronic liver diseases including Wilson’s disease or hemochromatosis.

### 2.7. Biochemical Tests

Blood tests were performed to determine levels of total cholesterol, low-density lipoprotein cholesterol (LDL-C), HDL-C, TG, aspartate aminotransferase (AST), alanine transaminase (ALT), gamma-glutamyl transferase (GGT), serum albumin, platelet count, glucose, and insulin levels.

### 2.8. Estimation Formulae for Hepatic Steatosis

Five relatively well-known estimation formulae were used for this study as shown below:Fatty liver index (FLI) [20] = e^x^/(1 + e^x^), where X = 0.953 × Log_e_(TG, mg/dL) + 0.139 × (body mass index (BMI), kg/m^2^) + 0.718 × Log_e_(GGT, U/L) + 0.053 × (waist circumference, cm) − 15.745;Hepatic steatosis index (HSI) [18] = 8 × ALT/AST ratio + BMI (+2 if type 2 diabetic; +2 if female);ZJU index [21] = BMI (kg/m^2^) + fasting plasma glucose (mmol/L) + TG (mmol/L) + 3 × ALT/AST ratio (+2 if female);NAFLD liver fat score (NAFLD-LFS) [17] = −2.89 + 1.18 × (metabolic syndrome—yes = 1, no = 0) + 0.45 × (type 2 diabetes—yes = 2, no = 0) + 0.15 × (fasting serum insulin, mU/L) + 0.04 × (AST, IU/L) − 0.94 × (AST/ALT);Framingham steatosis index (FSI) [22] = e^x^/(1 + e^x^), where X = −7.981 + 0.011 × age (years) − 0.146 × sex (female = 1, male = 0) + 0.173 × BMI (kg/m^2^) + 0.007 × TG (mg/dL) + 0.593 × hypertension (yes = 1, no = 0) + 0.789 × diabetes (yes = 1, no = 0) + 1.1 × ALT:AST ratio ≥ 1.33 (yes = 1, no = 0).

### 2.9. Statistical Analysis

Chi-square test was used to compare categorical variables according to the presence of fatty liver and Mann–Whitney U-test was used to compare continuous variables. The Kruskal–Wallis test and Jonckheere’s trend test were applied to analyze ordered differences in liver steatosis biomarkers between different steatosis grade groups. Post hoc analyses were performed to test differences between steatosis grades using the Mann–Whitney U-test with Bonferroni correction. To counteract the increasing familywise error rate of multiple comparisons, a Bonferroni correction of six was multiplied with original *p* values and adjusted *p* values were compared for significance. To assess the performance of the prediction formulae, areas under the receiver operator characteristics curves (AUROCs) were determined. In addition, we determined the sensitivity, specificity, PPV, and NPV at known cutoff points for each prediction model [17,18,20,21,22]. Pairwise comparisons between clinical formulae were performed using the method of DeLong et al. [23]. Statistically significant variables in univariate analysis were entered into a multiple logistic regression model to identify factors associated with the presence of NAFLD. We identified new optimal cutoff values for NAFLD-LFS using the Youden index. All statistical analyses were conducted using SPSS version 18.0 (SPSS Inc., Chicago, IL, USA) and MedCalc version 17.2 (MedCalc Software, Ostend, Belgium). A *p* value <0.05 was considered statistically significant.

## 3. Results

### 3.1. Baseline Characteristics

Baseline characteristics of study subjects according to hepatic steatosis status are summarized in Table 1 based. Among study subjects, 30.1% were diagnosed as having NAFLD by MRI (392/1301). This study included 1001 (76.9%) men. The mean age of all subjects was 50.7 ± 8.4 years. Prevalence of diabetes, hypertension, and metabolic syndrome were higher in the NAFLD group (all *p* < 0.05). Among patients diagnosed with NAFLD by MRI, 19.9% had diabetes, 38.5% had comorbid hypertension, and 22.2% had dyslipidemia. NAFLD patients accounted for 45.8% of diabetic patients, 35.6% of hypertension patients, and 43.2% of dyslipidemia patients.

### 3.2. Diagnostic Performance of Estimated Hepatic Steatosis Formulae

AUROCs of the following five estimation models were calculated for predicting fatty liver defined as liver fat >5% on MRI: FLI, HSI, ZJU index, NAFLD-LFS, and FSI (Table 2). All prediction models demonstrated fair diagnostic performances, with AUROCs ranging from 0.68 to 0.72. NAFLD-LFS demonstrated the highest diagnostic performance, with an AUROC value of 0.72, followed by FSI (0.70), HSI (0.69), ZJU index (0.69), and FLI (0.68) (Figure 2). NAFLD-LFS had a higher AUROC value than did FLI (*p* < 0.001), HSI (*p* = 0.005), FSI (*p* < 0.001), and ZJU index (*p* = 0.001) in pairwise comparison analyses. NAFLD-LFS demonstrated a sensitivity of 48.1%, a specificity of 83.4%, a PPV of 55.7%, and an NPV of 78.8% (Table 2). For males and females, the AUROCs of NAFLD-LFS were 0.73 (95% CI, 0.69–0.76) and 0.70 (95% CI, 0.63–0.78), respectively. The best cutoff value for predicting fatty liver was −1.33 in our sample. When applying the new cutoff value, NAFLD-LFS demonstrated a sensitivity of 63.5%, a specificity of 73.6%, a PPV of 51.0 %, and an NPV of 82.3 %.

### 3.3. Correlation between Hepatic Steatosis Estimation Formulae and Fatty Liver Grade

Scores of all steatosis estimation formulae were gradually increased in subjects with higher fatty liver grade determined by MRI (Figure 3). All formulae had significantly higher values for patients with steatosis than for those in the non-steatosis group, demonstrating that these indexes could diagnose the presence of steatosis (adjusted *p* < 0.001).

### 3.4. Comparison of Diagnostic Performance between NAFLD-LFS and USG

Of 1301 subjects eligible for this study, 549 (42.2%) were diagnosed with NAFLD by USG and 392 (30.1%) were diagnosed with NAFLD by MRI. For fatty liver diagnosis using ultrasound, its sensitivity, specificity, PPV, and NPV were 71.7% (281/392), 67.2% (549/817), 51.2% (281/549), and 83.2% (549/660), respectively. The diagnostic agreement for fatty liver was 70.9% between ultrasound and MRI and 72.7% between NAFLD-LFS and MRI (Figure 4).

### 3.5. Proportion of Gray Zone of Dual Cutoff Prediction Models

There were three types of prediction models using dual cutoff values. Low cutoff values of the three fatty liver prediction models showed acceptable NPVs (FLI: 82.3%; HSI: 82.5%; ZJU index: 83.6%). High cutoff value showed moderate PPVs (FLI: 47.2%; HSI: 50.9%; ZJU index: 50.9%; Table 2). There were considerable gray zones in the three fatty liver prediction models (Figure 5, FLI: 28.9%; HSI: 48.9%; ZJU index: 53.6%). When using the dual cutoff model (FLI, HIS, and ZJU index), 28.9~53.6% of subjects belonged to the gray zone. Thus, the prediction formula could not be applied.

A logistic regression analysis was performed to identify risk factors associated with the occurrence of hepatic steatosis. Platelet count, ALT/AST ratio, homeostatic model assessment of insulin resistance (HOMA-IR), and diabetes were found to be independent risk factors for fatty liver (Table 3). BMI and TG as, independent risk factors in FLI, HSI, and ZJU index hepatic steatosis prediction models were not independent risk factors for fatty liver in our multivariate analysis.

## 4. Discussion

This is the first study that validates previous fatty liver indices in a community-based population using MRI, “the next best method” as compared with liver biopsy. Among all five formulae, NAFLD-LFS had the best diagnostic performance (AUROC: 0.72), comparable to USG in a community-based population.

NAFLD-LFS was designed based on ^1^H-MRS [17]. Metabolic syndrome, type 2 diabetes, fasting insulin, AST, and ALT were included in this prediction model. The NAFLD-LFS AUROC in our study (AUROC: 0.72) was lower than that in the NAFLD-LFS original study population (AUROC: 0.87, sensitivity of 86% and specificity of 71%). This might be due to the fact that subjects included in the original study had moderate-to-high risk of fatty liver. NAFLD-LFS originated from the study enrolling subjects who were recruited to clinical studies via advertisements or referred to a tertiary medical institution because of chronically elevated serum transaminase concentrations. In the original study, prevalence rates of NAFLD, diabetes, and metabolic syndrome were 47%, 23%, and 57%, respectively. Subjects had an average BMI of 30.8 kg/m^2^ and a waist circumference of 104 cm. Our study differed from the original setting because data were gathered from subjects with average-to-low risk of metabolic syndrome.

In the present study, diagnostic performances of FLI (AUROC: 0.68) and HSI (AUROC: 0.69) widely used in real-world settings for steatosis calculation formulae were relatively low compared to their original description (AUROCs: 0.85 and 0.81, respectively) [18,20] and previous external validation studies (AUROC range: 0.78–0.87) [2,3,4,5,7,8,13,15,24]. Several factors might have contributed to such differences. First, both formulae were developed and validated using USG-based cohorts. Although USG is a method widely used in clinical practice, low diagnosis rates for mild steatosis and inter-/intra-observer errors are limitations of USG [25]. Second, the PPV critically depends on the prevalence of the disease in the study population [16]. Therefore, different study designs and NAFLD prevalences might cause different diagnostic performance. The development of FLI and HSI was based on case–control data sets that tended to include high proportions of NAFLD cases (40–50%). This suggests that the diagnostic performance in previous validation studies derived from cohorts with high fatty liver prevalence might have been overestimated [26]. In the present study, PPVs of the five indices were low (47.2–55.7%) despite high cutoff values. The prevalence of NAFLD was 30.1% in the current study, comparable to the prevalence of NAFLD in Asia (29.6%) reported in the most recently published meta-analysis study [27]. Owing to the low PPV, our results do not support the use of their applications as a general steatosis screening method at least in Asian populations having similar characteristics to those of the population used in the present study. Third, different characteristics of the study population can lead to different diagnostic performance. For example, BMI and waist circumference were risk factors in many prediction models. However, average BMI and waist circumference values are markedly different between Asian and Caucasian populations. In the present study, BMI and waist circumference were not factors significantly associated with hepatic steatosis in the multivariate analysis (Table 3).

Additionally, the three methods (FLI, HSI, and ZJU index) using dual cutoff values showed a wide gray zone (28.9–53.6%) where the diagnosis was undetermined (Figure 5). Of course, binary constraint of decision (with/without the disease) often does not fit the reality of clinical practice. However, when these scores are applied, it is very likely that more than a few subjects in the gray zone may undergo additional, more expensive, and invasive investigations. By contrast, NAFLD-LFS is used with a single cutoff without this limitation and it achieved comparable sensitivity and specificity results as those found in FLI, HSI, and ZJU index. Moreover, it showed the highest diagnostic performance, which was comparable to that of USG. Consequently, our results suggest that NAFLD-LFS can be used as a first-line screening tool for a large numbers of individuals.

This study has several limitations. First, the majority of subjects were middle-aged men because the study population comprised workers and their spouses who visited the health examination center. Since subjects visited the medical institution for health checkup purposes, the study group had a low risk of metabolic syndrome. However, the prevalence of diabetes and NAFLD did not differ significantly from that of the general population. Second, we defined fatty liver as the presence of hepatic fat fraction ≥5% on MRI. Unfortunately, values for diagnosing fatty liver disease are not yet conclusively defined. A study including 345 patients with no identifiable risk factors for hepatic steatosis has suggested that a 5.56% fat fraction is an abnormal level of hepatic fat using MRI-proton density fat fraction [28]. Although cutoff values proposed for the diagnosis of hepatic steatosis varied between 3.7% and 6.4% [9,19,29,30,31,32], based on the aforementioned study as a landmark, a fat fraction of 5% has generally been used to distinguish patients with fatty liver and those without fatty liver. Future studies correlating fat fraction, histology, and clinical outcomes may change this cutoff value. Finally, measurement of alcohol intake was based on participants’ self-reported questionnaires, which may lead to misclassification of study group. However, in our study, standardized questionnaires were used to quantify the amount of alcohol consumption addressing the quantity of different alcoholic drinks consumed during an average week. In conclusion, the NAFLD liver fat score showed the best diagnostic performance for detecting hepatic steatosis. Its diagnostic performance was comparable to that of USG in a community-based setting.

## Figures and Tables

**Figure 1 jcm-09-02851-f001:**
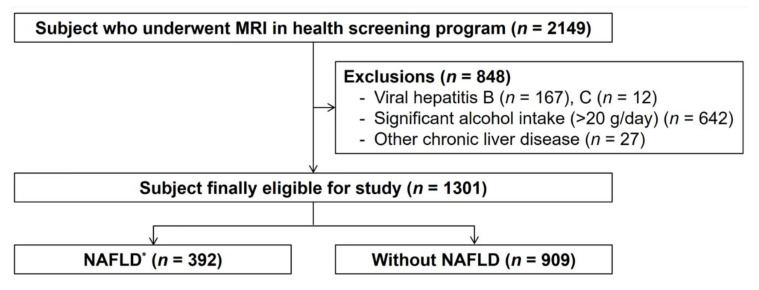
Flow chart showing enrollment of participants for this study. * NAFLD = Non-alcoholic fatty liver disease. Fatty liver was defined as hepatic fat fraction ≥5% on magnetic resonance imaging.

**Figure 2 jcm-09-02851-f002:**
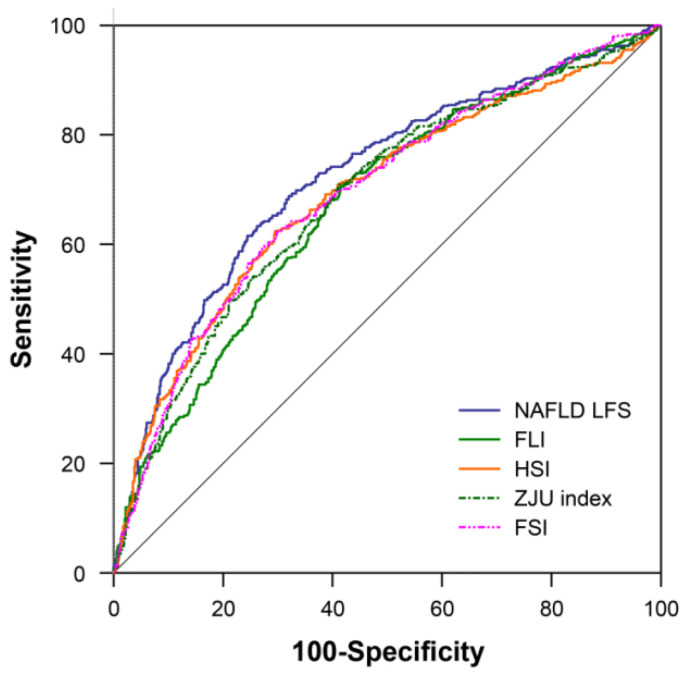
Area under the receiver operator characteristics (AUROC) of each prediction model for predicting fatty liver.

**Figure 3 jcm-09-02851-f003:**
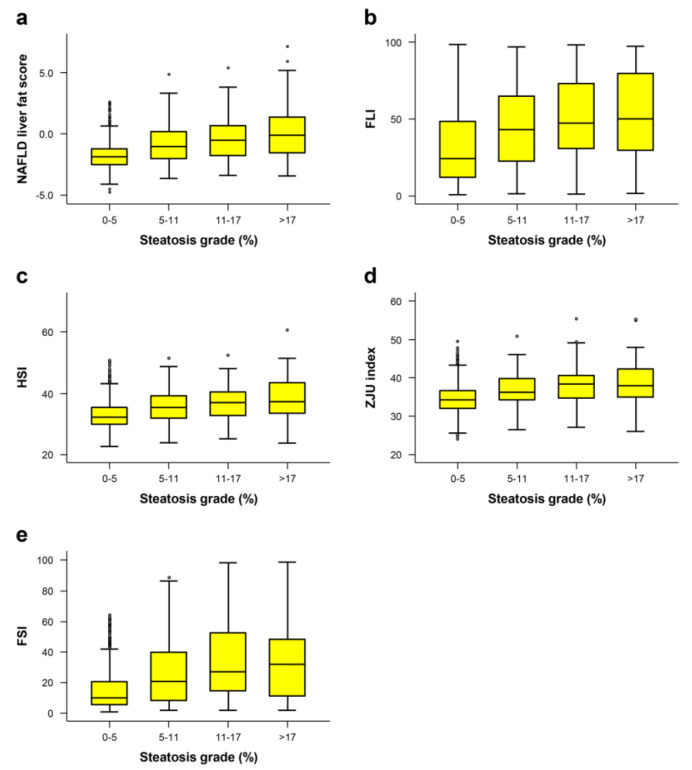
Box plot of each prediction model: (**a**) NAFLD liver fat score, (**b**) fatty liver index (FLI), (**c**) hepatic steatosis index (HSI), (**d**) ZJU index, and (**e**) Framingham steatosis index (FSI)) for predicting fatty liver according to the degree of hepatic steatosis. The yellow box represents the interquartile range and the black line across the box indicates the median. “Whiskers” are black lines that extend from the box to the highest and lowest values, excluding outliers (dots).

**Figure 4 jcm-09-02851-f004:**
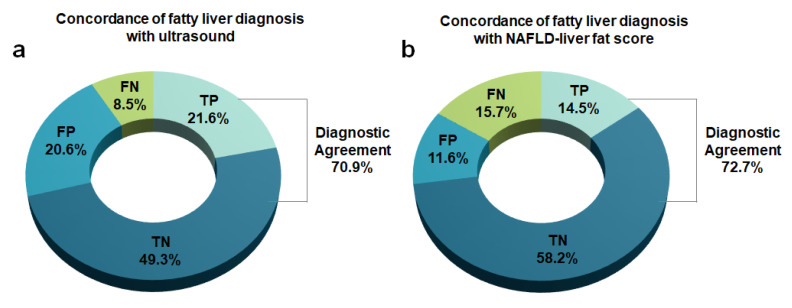
Concordance of fatty liver diagnosis with ultrasound (**a**) and non-alcoholic fatty liver disease liver fat score (**b**) in comparison with magnetic resonance imaging. TP: true-positive; TN: true-negative; FP: false-positive; and FN: false-negative.

**Figure 5 jcm-09-02851-f005:**
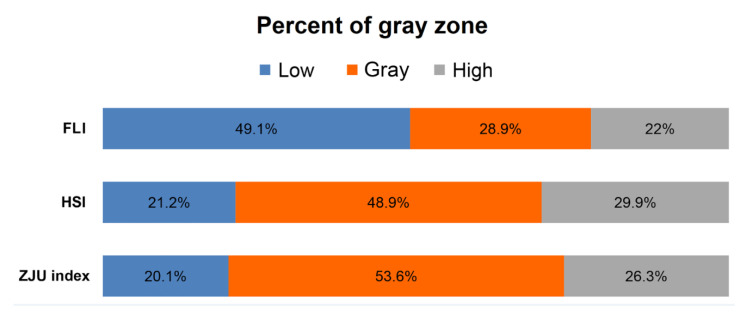
Proportions of gray zones among estimated formulae using two cutoffs.

**Table 1 jcm-09-02851-t001:** Baseline characteristics of the study population.

Characteristics	Total *n* = 1301 (100%)	No NAFLD *n* = 909 (69.9%)	NAFLD *n* = 392 (30.1%)	*p*-Value
Age (years)	51 (46–56)	51 (46–56)	50 (44–56)	0.013
Males, *n* (%)	1001 (76.9%)	676 (74.4%)	325 (82.9%)	0.001
Body mass index (kg/m^2^)	24.4 (22.6–26.2)	24 (22.3–25.6)	25.6 (23.6–27.5)	<0.001
Waist circumference (cm)	85.3 (80.0–90.8)	84 (79.0–89.5)	88.6 (83.0–93.5)	<0.001
Diabetes, *n* (%)	170 (13.1%)	92 (10.1%)	78 (19.9%)	<0.001
Hypertension, *n* (%)	423 (32.5%)	272 (29.9%)	151 (38.5%)	0.002
Metabolic syndrome (%)	321 (24.7%)	164 (18.0%)	157 (40.1%)	<0.001
Platelet (10^9^/L)	237.0 (206.0–272.0)	234.0 (204.0–268.0)	248 (213.5–279.0)	0.001
AST (U/L)	22.0 (19.0–28.0)	21.0 (18.0–26.0)	25.0 (21.0–34.0)	<0.001
ALT (U/L)	23.0 (17.0–34.0)	21.0 (16.0–28.0)	32.0 (22.0–47.5)	<0.001
ALT/AST ratio	1.0 (0.8–1.3)	1.0 (0.8–1.2)	1.2 (0.9–1.5)	<0.001
GGT (U/L)	29.0 (19.0–52.0)	26.0 (17.0–44.0)	39.0 (24.0–67.5)	<0.001
TG (mg/dL)	111.0 (79.0–162.0)	102.0 (74.0–149.0)	132.5 (92.5–199.0)	<0.001
HDL-C (mg/dL)	54.0 (45.0–65.0)	56.0 (47.0–66.0)	50.0 (42.0–60.0)	<0.001
LDL-C (mg/dL)	132.0 (108.0–154.0)	130.0 (106.0–151.0)	138.0 (111.5–160.0)	0.001
Fasting glucose (mg/dL)	98.0 (92.0–105.0)	97.0 (91.0–104.0)	99.5 (93.5–110.0)	<0.001
HbA1c (mmol/L)	5.6 (5.4–5.8)	5.6 (5.4–5.8)	5.7 (5.5–5.9)	<0.001
FLI	30.9 (14.1–55.8)	24.3 (12.2–48.6)	46.2 (25.6–71.2)	<0.001
HSI	33.4 (30.5–37.1)	32.5 (30–35.3)	36.1 (32.6–40.4)	<0.001
ZJU index	34.9 (32.5–38.1)	34.2 (32.2–36.7)	37.2 (34.6–40.2)	<0.001
NAFLD-LFS	−1.8 (−2.4–−0.6)	−2.0 (−2.5 – −1.2)	–0.7 (−1.9–0.5)	<0.001
FSI	12.4 (6.5–29.3)	9.9 (5.7–20.4)	24.6 (10–46.1)	<0.001
Fatty liver on USG (%)	549 (42.2%)	268 (29.5%)	281 (71.7%)	<0.001

Data are represented as median (25th–75th percentile) or as frequency (percentage). ALT = alanine transaminase, AST = aspartate transaminase, FLI = Fatty Liver Index, FSI = Framingham Steatosis Index, GGT = gamma-glutamyl transferase, HbA1c = hemoglobin A1c, HDL-C = high-density lipoprotein cholesterol, HSI = Hepatic Steatosis Index, LDL-C = low-density lipoprotein cholesterol, MRE = magnetic resonance elastography, NAFLD = non-alcoholic fatty liver disease, NAFLD-LFS = non-alcoholic fatty liver disease liver fat score, TG = triglyceride, USG = ultrasonography.

**Table 2 jcm-09-02851-t002:** Diagnostic performance of prediction models for fatty liver compared to magnetic resonance imaging (MRI).

Prediction Models	FLI		HSI		ZJU Index		NAFLD-LFS		FSI
AUROC (95% CI)	0.68 (0.64–0.71)		0.69 (0.66–0.73)		0.69 (0.66–0.72)		0.72 (0.69–0.75)		0.70 (0.66–0.73)
Cutoff value	Low	High	Low	High	Low	High	Traditional	New	
	(30)	(60)	(30)	(36)	(32)	(38)	(−0.64)	(−1.33)	(23)
Sensitivity (%)	71.2	34.4	87.8	50.5	89.0	44.4	48.1	63.5	52.0
Specificity (%)	57.8	83.4	25.0	79.0	24.1	81.5	83.4	73.6	77.1
PPV (%)	42.1	47.2	33.5	50.9	33.6	50.9	55.7	51.0	49.5
NPV (%)	82.3	74.7	82.5	78.7	83.6	77.3	78.8	82.3	78.9

AUROC = area under the receiver-operating-characteristic curve; FLI = Fatty Liver Index; FSI = Framingham Steatosis Index; HSI = Hepatic Steatosis Index; NAFLD-LFS = non-alcoholic fatty liver disease liver fat score; NPV = negative predictive value; PPV = positive predictive value.

**Table 3 jcm-09-02851-t003:** Risk factors associated with hepatic steatosis.

	Univariable			Multivariable		
Variables	OR	95% CI	*p*-Value	OR	95% CI	*p*-Value
Male (%)	1.67	(1.24–2.26)	0.001			
Age (years)	0.98	(0.97–1.00)	0.007			
BMI (kg/m^2^)	1.18	(1.14–1.23)	<0.001			
Waist Circumference (cm)	1.06	(1.04–1.07)	<0.001			
Platelet (10^9^/L)	1.00	(1.00–1.01)	0.002	1.00	(1.00–1.01)	0.009
ALT/AST	4.73	(3.42–6.53)	<0.001	2.49	(1.70–3.65)	<0.001
GGT (U/L)	1.01	(1.00–1.01)	<0.001			
TG (mg/dL)	1.00	(1.00–1.00)	<0.001			
HDL-C (mg/dL)	0.98	(0.97–0.99)	<0.001			
HOMA-IR	1.48	(1.35–1.63)	<0.001	1.17	(1.05–1.30)	0.006
Diabetes	2.21	(1.59–3.06)	<0.001	1.55	(1.06–2.26)	0.023
Hypertension	1.47	(1.15–1.88)	0.002			

AST = aspartate aminotransferase; BMI = body mass index; GGT = gamma glutamyl transpeptidase; HDL-C = high density lipoprotein cholesterol; HOMA-IR = homeostasis model assessment of insulin resistance, TG = triglyceride.

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
