# Peer review of "Comparative Assessment and External Validation of Hepatic Steatosis Formulae in a Community-Based Setting"

_jcm, 2020, doi:10.3390/jcm9092851_

Round 1

Reviewer 1 Report

The current manuscript validates, with a retrospective study design, previously described fatty liver indices to diagnose and detect hepatic steatosis in a community-based population using MRI based analysis. The primary aim of the study was to validate hepatic steatosis formulae using MRI with diagnostic performance assessed using the area under the ROC. Results of the analysis showed that the NAFLD liver fat score was the best diagnostic feature for detecting hepatic steatosis. This feature was comparable to USG measurements in community-based settings. While the findings and study design are elucidated with scientific rigor and robust experimental design, there are some minor concerns that need addressed:

  1. Did the authors include any retrospective data from females? The study design seems biased towards the male gender. Clarification on this would help the audience and would add to the usability of the model, not limitation.
  2. The exclusion criteria include alcohol consumption >20g/day. Could the authors comment on the cut off? Also, how reliable was the self-assessment and determination of the dose per day based on the questionnaire?
  3. Please provide a description on the exclusion criteria used for other chronic liver diseases. Did the authors exclude any/all individuals with PSC, PBC, AILD, AIH?
  4. The reviewer would also urge the authors to perform a thorough proofread and check for grammatical errors. Language editing is highly recommended and necessary for revision of the manuscript.
  5. In the three types of prediction models using dual cut-off values, the authors found the existence of considerable grey zones with the lowest in NAFLD-LFS model. However, 28.9% of the patients still remain clustered in this indeterminate grey zone potentially risking and sometimes unnecessary invasive and expensive diagnostic investigations. Can the authors comment how this issue can be addressed and (potentially) resolved? Can data from other risk factors like HOMA-IR and/or platelet counts be included in the formulae? Or use any other clinical feature that would better the outcomes using NAFLD-LS scores?

Author Response

Reviewer 1

Comments and Suggestions for Authors

The current manuscript validates, with a retrospective study design, previously described fatty liver indices to diagnose and detect hepatic steatosis in a community-based population using MRI based analysis. The primary aim of the study was to validate hepatic steatosis formulae using MRI with diagnostic performance assessed using the area under the ROC. Results of the analysis showed that the NAFLD liver fat score was the best diagnostic feature for detecting hepatic steatosis. This feature was comparable to USG measurements in community-based settings. While the findings and study design are elucidated with scientific rigor and robust experimental design, there are some minor concerns that need addressed:

  1. Did the authors include any retrospective data from females? The study design seems biased towards the male gender. Clarification on this would help the audience and would add to the usability of the model, not limitation.

Author responses: Thank you for your comment. We compared the diagnostic performance of five clinically widely used fatty liver prediction models (FLI, HSI, ZJU, NAFLD-LFS, and FSI) in a community-based cohort with average risk. In this study cohort, NAFLD-LFS showed the best diagnostic performance among the five prediction models (AUROC: 0.72, 95% CI: 0.69-0.75). As reviewer pointed out, in our cohort, the proportion of males was relatively higher than that of females due to the characteristics of the health examination center. When our cohort was divided into male and female groups, NAFLD-LFS showed similar diagnostic performance in both male and female groups (AUROC: 0.73 (95% CI: 0.69-0.76) in men, 0.70 (95% CI: 0.63-0.78) in women). In addition, NAFLD-LFS showed similar or superior diagnostic performance compared to the other four prediction models (FLI, HSI, ZJU, and FSI) in both male and female groups. We have described about this in result (p.6, line: 184-185).

  1. The exclusion criteria include alcohol consumption >20g/day. Could the authors comment on the cut off? Also, how reliable was the self-assessment and determination of the dose per day based on the questionnaire?

Author responses: Thank you for your comment. In previous studies for development of HSI and NAFLD-LFS, they set maximum acceptable alcohol intake to 20g/day for both men and women, which is below the level set of AASLD guideline (>210g/week for men and >140g/week for women) or WHO (harmful drinking, >60g/day for men and >40g/women). We adopted stricter standard to exclude alcohol interference as much as possible. We added these two references (Ref. Digestive and Liver Disease 42 (2010) 503–508, Gastroenterology 2009;137:865–872) in the methods section (p.2, line: 76, Ref 17,18).

Because alcohol intake was assessed using self-administered questionnaires, self-reporting bias, such as social desirability or recall bias, was inevitable. However, so far, there’s no other way to find out amount of alcohol consumption objectively. In our study, standardized questionnaires were used to quantify the amount of alcohol consumption addressing the quantity of different alcoholic drinks consumed during an average week. We have described about this as a limitation of our study (p.11, line: 283-287).

  1. Please provide a description on the exclusion criteria used for other chronic liver diseases. Did the authors exclude any/all individuals with PSC, PBC, AILD, AIH?

Author responses: Thank you for your comment. Participants with suspected chronic liver disease or liver cirrhosis on ultrasonography were excluded from the analysis (n=27) in order to exclude chronic liver disease with unknown etiology such as autoimmune hepatitis, primary biliary cirrhosis, and hemochromatosis. We added this sentence in the method section (p.2. line: 77-79).

  1. The reviewer would also urge the authors to perform a thorough proofread and check for grammatical errors. Language editing is highly recommended and necessary for revision of the manuscript.

Author responses: Thank you for our comment. We performed proofreading.

  1. In the three types of prediction models using dual cut-off values, the authors found the existence of considerable grey zones with the lowest in NAFLD-LFS model. However, 28.9% of the patients still remain clustered in this indeterminate grey zone potentially risking and sometimes unnecessary invasive and expensive diagnostic investigations. Can the authors comment how this issue can be addressed and (potentially) resolved? Can data from other risk factors like HOMA-IR and/or platelet counts be included in the formulae? Or use any other clinical feature that would better the outcomes using NAFLD-LS scores?

Author responses: We sincerely apologize for confusing you. There was a typographic error at figure 5 and we changed “NAFLD-LFS” to “FLI”.

Three types of prediction models using dual cut-off (FLI, HSI, ZJU index) leaved considerable grey zones (28.9%~53.6%) where the diagnosis is undetermined. By contrast, NAFLD-LFS is used with a single cut-off without this limitation and it achieved comparable sensitivity and specificity results as those found in FLI, HSI and ZJU index. Moreover, it showed the highest diagnostic performance which was comparable to that of ultrasonography. Consequently, our results suggest that NAFLD-LFS can be used as a first-line screening tool for a large numbers of individuals. We have additionally described about this at discussion section (p.11, line: 283-287)

Reviewer 2 Report

Dear authors thank you very much for submitting this paper to the journal

The main limitations of the study are the focus on steatosis grade which is not significant clinically (fibrosis stage is the main predictor in NAFLD). Plus, the percentage of patients falling in the grey areas is quite significant.

As there is no evidence supporting the screening for Nafld in the general population, it might be worth focusing on subgroup at higher risk (diabetics, obese, metabolic syndrome). 

Author Response

Reviewer 2

Comments and Suggestions for Authors

Dear authors thank you very much for submitting this paper to the journal

The main limitations of the study are the focus on steatosis grade which is not significant clinically (fibrosis stage is the main predictor in NAFLD). Plus, the percentage of patients falling in the grey areas is quite significant.

As there is no evidence supporting the screening for Nafld in the general population, it might be worth focusing on subgroup at higher risk (diabetics, obese, metabolic syndrome). 

Author responses: Thank you for your comment. We totally agree that predicting fibrosis stage is more clinically significant than that of fatty liver. A very recent work showed that 6-7% of the adult population without known liver disease have liver fibrosis and suggested that screening for liver fibrosis in the general population should be assessed (Ref. Lancet Gastroenterol Hepatol. 2016;1:256-60). In that study, liver fibrosis was mostly associated with NAFLD. Therefore, in the first instance, diagnosis of NAFLD should be performed to identify patients at high risk of hepatic fibrosis. Although hepatic fibrosis is a more serious and clinically significant disorder, early detection of simple steatosis is also an important and promising field from a public health system. We additionally described this in introduction (p. 1, line: 35-37)

Round 2

Reviewer 2 Report

no comments